# Food Waste Management with Technological Platforms: Evidence from Indian Food Supply Chains

**Youthika Chauhan**

Kenan-Flagler Business School, University of North Carolina, Chapel Hill, NC 27599, USA;
Youthika_Chauhan@kenan-flagler.unc.edu; Tel.: +1-919-638-4175

**Abstract:** Feeding the people sustainably continues to be a challenge in the present times. Enormous amounts of food wastage aggravate this problem. In developing countries, food wastage primarily occurs within the supply chain. Lack of technological infrastructure in these countries causes significant post-harvest loss. While research shows that developments in food supply chains can reduce food wastage, no systematic research has been done so far to show the possible relationship between the use of technology and food loss. This paper attempts to address this gap by studying the supply chains of different food processing organizations in India to assess the role of technological platforms in reducing food wastage in supply chains. Using a qualitative inductive methodology, the author identified the technological platforms that can address food wastage. Then, using multiple case-study analysis, the supply chains of sample firms were evaluated. The author assessed the food loss in these supply chains through comparative analysis to draw conclusions about the effectiveness of selected technological platforms. This study provides managers in the food industry with insights to prevent food loss, as well as some policy implications for developing economies. Overall, this paper throws light on the issue of food wastage and the possible means for its prevention.

**Keywords:** food systems; supply chain; technology; food wastage; sustainability; multiple-case design

## 1. Introduction

The food and agricultural systems of the world have been feeding more people than before. However, although more food is being produced, the problems of hunger and nutrient deficiencies are prevalent [1]. Aggravating this problem is the fact that about fourteen percent of food produced globally is lost during the post-harvest production stage [1,2]. In other words, 1.4 Gt of food suitable for human consumption is wasted each year [3,4].

Although in developed countries, food wastage mainly occurs at the consumer end, in developing countries, food wastage primarily occurs within the supply chain [5]. As this paper focuses on the ways to reduce food wastage in supply chains, the author limits the discussion to the food wastage in developing economies where the lack of infrastructure is the key reason for significant post-harvest loss [2,6]. Studies shows that post-harvest to distribution loss is highest in central and southern Asia, at nearly 21% [1,7]. Further, 85% to 90% of the observation points in central and southern Asia are from India, suggesting that food loss in supply chains are a major problem in the country. Indeed, several sources state that nearly 40% of the food produced in India is wasted [8–11].

A range of factors, such as microbial, enzymatic, chemical, physical, and mechanical ones, lead to food spoilage [12,13]. These factors necessitate the development of logistics systems in food supply chains [14]. Computerization and technological platforms facilitating online communications within food supply chains can facilitate the management of agricultural resources [15]. Researchers suggest that supply chains with advanced technological platforms can prevent nearly 50% of such loss [16].

These findings emphasize the potential of food supply chains to reduce food loss and achieve higher food security.

Although literature shows that technological developments have the potential to reduce food wastage in supply chains [16], research does not clearly show the possible relationship between the use of technology and reduced food loss. The aim of this paper is to address this gap by studying the supply chains of different organizations and their food wastage. Specifically, this study aims to answer the research question: *whether, and if yes, under what conditions, can technological platforms help reduce food wastage in supply chains?* Overall, by addressing this gap about food wastage in supply chains, this paper describes the possible transformation of the existing food supply chains into more sustainable ones for the future.

This paper is structured as follows: The remainder of Section 1 describes the conceptual framework and the technological platforms studied in this paper. Section 2 reports the research method, sample, and data analysis. Section 3 reports the results of the analysis. Sections 4 and 5 refer to the discussion and policy implications, respectively, along with some concluding remarks.

## 1.1. Conceptual Framework

The conceptual framework of this paper includes an explanation of what a food supply chain is, and then explains how the author studies these supply chains to assess their wastage. The author uses the food systems (approach as described by the United Nations High-Level Panel of Experts on Food Security and Nutrition; [17]) to define the food supply chains and further identify opportunities for reducing food wastage in the food processing industry in India. According to the HLPE United Nations (2017) report, the food supply chain consists of the activities and actors that take food from production to consumption as well as the disposal of its waste [18]. To study food supply chains, the author first identifies a suitable context (i.e., India, for the reasons described earlier). This is followed by sample selection and identification of a suitable methodology. Then, the author identifies the different drivers of food wastage, and the technologies that can help address it.

First, different organizations are identified for multiple case-study analysis [19]. To make the sample representative, the author selected firms of different sizes, ranging from entrepreneurial firms with <10 employees to subsidiaries of multi-national companies having >1000 employees. The firms also belong to different industry segments such as dairy, frozen foods, and confectionery, which gives a holistic picture of the industry. The differences in the supply chain have been taken into account during the analysis. As described in the Section 2.1 Sample Overview, these firms are selected from different parts of India. The author continued collecting data from firms of different sizes, from different industry segments and locations, until she attained theoretical saturation, i.e., a point where no additional insights were obtained upon collecting further data [19,20]. Collecting data from seventeen firms allowed for a significant amount of diversity within the supply chains, as well as the attainment of theoretical saturation. Moreover, seventeen is within the range suggested by Eisenhardt (1989) for the number of cases in multiple case-study analysis [19].

With the help of interviews with firm employees in the supply chains, the author prepared case studies for each firm. This was followed by an investigation into the vulnerabilities in the supply chains that might lead to food wastage. Then, the author assessed the technological infrastructure of these firms. Depending on the vulnerabilities in the supply chains, the role of technology platforms in addressing food wastage was evaluated.

## 1.2. Technological Platforms

The author used the key sources of literature in supply chain management, production technologies, and operations management to identify the most important and relevant technological platforms [21–24]. The author studied their role in the supply chains and evaluated their relevance through some initial discussion with experts from the food industry (i.e., interviewees from C1, C2, C4, C7, C9, and C14). The role of these technologies was also discussed with experts from academia (i.e., two professors of

supply chain management, and one lecturer in food technology and engineering). Based on these insights and from prior literature, the author identified specific technological platforms that can improve the efficiencies in food supply chains:

i.      Internet-based data monitoring and communication.
ii.     Enterprise Resource Planning (ERP), i.e., software that helps integrate components of a company, including supply chain, by sharing and organizing information among participants at different levels [23].
iii.    Supply Chain Event Management (SCEM): this term refers to methods that process supply chain events [25]. In other words, it is a process of monitoring the planned sequence of activities systems along a supply chain and reporting any errors with the help of computerized monitoring devices [26].
iv.     Radio Frequency Identification (RFID) systems, i.e., small electronic tags that track the position and movement of items [27,28].
v.      Electronic Data Interchange (EDI), i.e., computer-to-computer exchange of documents for order processing, transactions, accounting, production, and distribution [23].
vi.     Programmable Logic Controller (PLC), i.e., a control system to monitor parameters of input devices and to generate decisions-based output parameters [23].
vii.    Cloud computing, i.e., an internet-based system to access a shared pool of computing resources (Mell & Grance, 2011).
viii.   Machine-to-machine, i.e., M2M communication or wireless or wired technology that captures data from a remote location using sensors and connects to the back-end enterprise systems via WLAN, satellite, or cellular communication [29,30].

In addition to the aforementioned technological platforms, literature described several other terms referring to the application of these platforms, such as logistics execution systems, network design applications, warehouse and transportation planning systems, and dashboard analytics for display and monitoring. This study includes these applications as well in order to have a holistic understanding of food supply chains and their technological infrastructure.

The conceptual framework for this study is described in the flowchart below (Figure 1):

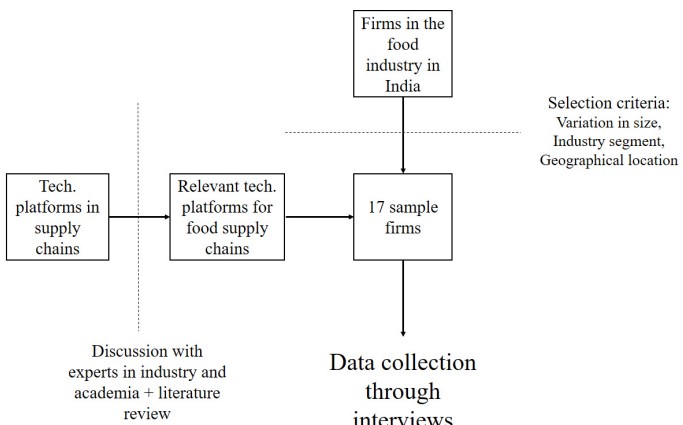

**Figure 1.** Conceptual framework.

This figure describes the considerations and steps leading to the data collection.

## 2. Materials and Methods

This paper uses qualitative analysis because qualitative study lends itself well to answer open ended questions such as the one studied in this paper [31]. Specifically, this paper uses qualitative inductive methodology [32,33] described by Gioia, Corley, and Hamilton (2013) [34]. This method

captures the concepts relevant to organizational processes in an inductive manner, using grounded theory [20] building on procedures for open-ended inductive theory building [35]. In the early stage of the analysis, the author identified a number of informant terms, codes, and categories. Moving forward, in the second order analysis, the author sought similarities and differences among the aforementioned categories. Wherever the author found concepts that were repeatedly present [36], these concepts were put together in the same themes. Finally, from these themes, broader dimensions related to the causes related to food wastage in supply chains (e.g., perishability, supply chain complexity) were identified. The author iterated between the data and literature several times, as suggested in the literature [36], in order to identify the relevant factors. Additional data was collected until a point of "theoretical saturation" was attained [20]. Prior literature was then used to explain these concepts and to use them further for analysis. This inductive analysis was followed by comparative analysis of the firms. Relevant parameters were rated in a comparative manner. Thereafter, the author formulated propositions based on the prior literature, and assessed the propositions on the firms. These steps are elaborated in the subsequent sections.

*2.1. Sample Overview*

In developing economies, food wastage primarily occurs in supply chains [5]. More specifically, post-harvest to distribution loss is highest in central and southern Asia [7], and significant in India [8,11], as described earlier. Hence, food supply chains in India were selected for this this study. To create a representative sample, the author chose organizations from different segments within the food industry and various parts of India. Their supply chains vary in complexity, volume, nature of the products, and market demands. Thus, the organizations are selected in a manner that takes their diversity and representativeness into account. For each firm, the author interviewed an employee who was closely involved with the products' supply chain. The interview questions are described in Appendix A. Table 1 below describes the studied sample. All company names have been disguised to maintain anonymity.

The Figure 2 below describes the location of each of the sample firms and shows their geographical spread.

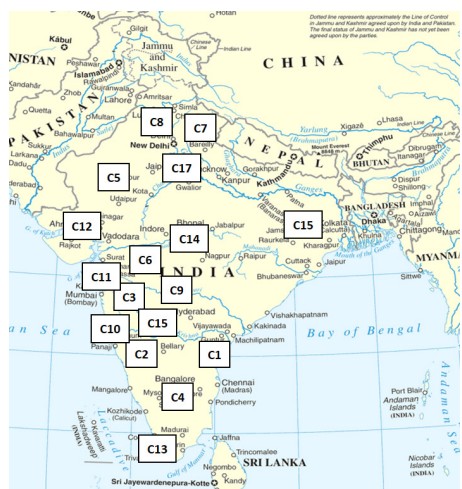

**Figure 2.** Geographical location of sample firms.

**Table 1.** Sample description.

| Firm | Company Description | Respondent | Location |
|------|---------------------|------------|----------|
| C1 | Indian FMCG (fast-moving consumer goods) conglomerate. Key products: wheat flour and snacks. | Sales manager | Tamil Nadu |
| C2 | A subsidiary of a European FMCG giant. Key product: instant noodles. | Quality executive | Goa |
| C3 | A subsidiary of an American MNC (multi-national company). Key product: tomato ketchup. | R&D manager | Maharashtra |
| C4 | A subsidiary of an American MNC. Key products: pizza and desserts. | Production manager | Karnataka |
| C5 | A subsidiary of a European brewery. Key product: beer. | Quality executive | Rajasthan |
| C6 | A subsidiary of a European FMCG giant. Key product: chocolate and other confectioneries. | R&D manager | Maharashtra |
| C7 | A subsidiary of European FMCG. Key product: candy and other confectioneries firm. | Trade manager | Haryana |
| C8 | Traditional snack-foods manufacturer. Key product: snack foods, meals for same-day consumption. | Quality auditor | Haryana |
| C9 | Indian dairy company. Key product: ice-cream. | Quality executive | Maharashtra |
| C10 | Subsidiary for European flavor manufacturer. Key products: flavors for industrial use. | Production executive | Maharashtra |
| C11 | Indian conglomerate. Key products: tea, coffee, and soup premixes. | Supply chain manager | Maharashtra |
| C12 | A subsidiary of a Canadian MNC. Key products: frozen snack foods. | Quality executive | Gujarat |
| C13 | Indian spice manufacturer and exporter. Key products: spices, spice blends, and extracts. | Production head | Kerala |
| C14 | Indian nutraceuticals (nutraceutical is a substance that may be considered food and provides medical or health benefits, including the prevention and treatment of disease [37]) for dietary supplements (a dietary supplement is a product containing a "dietary ingredient" intended to add further nutritional value to (supplement) the diet. A "dietary ingredient" may be one or a combination of substances [38]). Key products: lutein extract, capsaicin extract. | Production executive | Maharashtra |
| C15 | Contract manufacturing unit for an Indian FMCG manufacturer. Key product: biscuits. | Quality executive | Maharashtra |
| C16 | Indian manufacturer of traditional remedial formulations sold in the retail market. Key products: dietary supplements. | Production manager | West Bengal |
| C17 | Manufacturer to several prominent companies in the Indian healthcare industry. Key products: Indian traditional dietary supplements. | Senior manager | Delhi |

Although the figure shows that a large number of sample firms are present in western India, it is an indicator of the representativeness of the sample. This is because western parts of India are more industrialized than the eastern region [39,40]. Consequently, there are more food processing firms and supply chains in western India as compared to the east.

After creating the sample of firms, the author proceeded with the data collection.

## 2.2. Data

The author took semi-structured interviews based on a questionnaire, allowing individual responses to guide further questions. The questions pertained to the following areas: the product and its perishability, technological infrastructure in the supply chain, and food wastage. All interviews were in English. The author recorded each interview with permission from the respondent. Each interview lasted between 45 min and 1.5 h, with 75 min being the average interview duration. In total, 21.5 h of interview data was collected. The entire process of data collection lasted four months. Thereafter, the interviews were transcribed and used for analysis. Wherever required, the author collected additional information via additional discussions and email communication.

The author supplemented the primary data comprising of interview notes with secondary archival information available from the website of each company. This was followed by case-study writing for each firm. Wherever required, additional information from the interviewees was sought over email.

## 2.3. Analysis

The interview data suggested that food loss can occur at several stages in the supply chain, including raw material procurement, storage, production, dispatch, logistics, and retail. Moreover, different products have different requirements for processing and storage that may demand different forms of technologies. Based on preliminary interviews and prior literature, the author identified factors that determine the need for technological platforms in the supply chain to prevent food loss [12,41]. These factors are:

i. Supply chain complexity: Milgate (2001) [41] describes supply chain complexity as the uncertainty, technological intricacy, and organizational systems required to manage it. In other words, supply chain complexity refers to the number of production processes and needs for stringent control of processing and storage conditions [12,15]. From the interview responses, the author evaluated how complex the supply chains of each organization was. The author rated the sample firms on these parameters on three levels. For example, if the uncertainty in the processes (e.g., seasonality, reliance on weather), technological intricacy (need for advanced technology), and requirement of organizational systems were high (e.g., manual monitoring and supervision), the author classified supply chains as highly complex and rated as "5". This took into account the number of production processes and need for stringent control for processing and storage conditions. Similarly, if the processes did not have a high level of uncertainty, technological intricacy, and need for organizational systems, the supply chain was classified as "moderately complex" and rated as "3". Finally, if the processes had very low uncertainty, technological intricacy, and need for organizational systems, the supply chains were classified as "less complex" and rated as "1". Supply chains falling between these three levels were rated at 4 and 2, respectively. These ratings allowed us to evaluate the supply chains in a comparative manner and study the differences.

ii. Perishability of raw material and product. According to the US Department of Agriculture [42], there is a likelihood for food to spoil, decay, or become unsafe for consumption if not maintained at specific conditions. Rahman (2005) and Singh and Heldman (2001) describe the criteria for the perishability of food products [12,43]. Based on these explanations, and based on the responses about the perishability of the raw material and final product as described by our respondents, they gave the following ratings:

1. Perishable: shelf-life of 1–2 days, rated 5.
2. Semi-perishable: shelf-life of up to 1–2 weeks, rated 4.
3. Less perishable: shelf-life of 3–4 weeks, rated 3.
4. Shelf-stable: shelf-life of >1 month, rated 2.
5. Non-perishable: shelf-life of 12 months, rated 1 (Rahman, 2005).

As understood from literature and interviews, these two parameters, i.e., supply chain complexity and perishability, largely determine the levels of food wastage when technological deployment is not taken into account. For example, high supply chain complexity can cause higher chances of operational errors that can lead to process failures and food loss. Additionally, high perishability makes raw materials or products vulnerable to spoilage easily when storage and processing conditions deviate from the required conditions. These two factors, i.e., supply chain complexity and perishability, were combined into another parameter, production sensitivity (PS). The author calculated production sensitivity as the sum total of "supply chain complexity" and "production sensitivity". Thus, the production sensitivity of the supply chains was rated on a scale of two to 10. Further, to analyze the firms in a comparative manner, the author classified them as either (i) having high production sensitivity if the production sensitivity rating is between 2 and 6, or (ii) having low production sensitivity, if the production sensitivity rating is between 7 and 10. This was followed by an assessment of the levels of food wastage.

All interviewees described food wastage in the supply chain as a percentage of production volume. The values of food wastage across all the sample firms ranged from ~0% to 10%. Hence, food wastage levels were classified as follows:

i.　Low: < 2% of production volume.
ii.　Medium-low: 2–4% of production volume.
iii.　Medium: 4–6% of production volume.
iv.　Medium-high: 6–9%.
v.　High: > 9%.

The author also rated the firms according to the technology deployed at each organization. The author asked the interviewees whether the technological platforms described in Section 1.2 were being used at the organization or not. Table 2 presents the responses from each firm, and the corresponding rating. The same labels from Section 1.2 are used to refer to the technological platforms. Y denotes the presence of the technological platform while N denotes its absence. O denotes the presence of the technology at other locations. The presence of every technology contributed to 1 point in the rating, and the presence in another location contributed to 0.5 as it showed that the supply chain had technology, although it may not be entirely useful for the supply chain. The author gauged this qualitatively from the interview data. TR refers to the technology rating.

**Table 2.** Technological deployment in the sample firms.

|  | i | ii | iii | iv | v | vi | vii | viii | Comment | TR |
|---|---|---|---|---|---|---|---|---|---|---|
| C1 | Y | Y | Y | Y | N | Y | N | N | PLC for automation, backward integrated | 7 |
| C2 | Y | Y | Y | N | Y | O | O | N | PLC for automation | 5 |
| C3 | Y | N | Y | Y | Y | N | O | N |  | 4.5 |
| C4 | Y | Y | N | Y | O | N | O | N | Air humidifiers are automated | 4 |
| C5 | Y | Y | Y | O | Y | N | Y | O | PLC present but not for automation | 6 |
| C6 | Y | Y | N | Y | N | N | N | N | PLC present but not for automation | 3 |
| C7 | Y | Y | Y | N | Y | N | N | N | PLC for automation | 4 |
| C8 | Y | N | Y | N | N | N | N | N | Barcode used until central location | 2 |
| C9 | Y | N | N | N | N | N | N | N | No ERP until franchisee location. Used thereafter | 1.5 |
| C10 | Y | N | O | O | N | O | N | N | ERP integrated with the suppliers | 2.5 |
| C11 | Y | N | Y | N | N | N | N | N |  | 2 |
| C12 | Y | Y | Y | O | O | O | O | N | Air humidifiers automated by tech. platforms | 5 |
| C13 | Y | N | N | N | N | N | N | N |  | 1 |
| C14 | Y | N | N | N | N | N | N | N |  | 1 |
| C15 | Y | Y | Y | N | N | N | N | N | PLC present but not for automation | 3 |
| C16 | Y | N | Y | N | N | N | N | N |  | 2 |
| C17 | N | N | N | N | N | N | N | N |  | 0 |

Finally, using production sensitivity (PS), the author evaluated the food wastage levels vis-à-vis the technology deployed at these organizations. From prior literature [12,43], the author found that when PS is high, supply chains require high technology deployment to prevent food wastage. If the production sensitivity is low, food wastage may be moderate or low even in the absence of technological deployment. These broad level understandings were tested on the data through the propositions presented in Table 3 below.

**Table 3.** Propositions on expected food wastage level.

| Proposition | Production Sensitivity | Technology Deployment | Expected Food Wastage |
|:---:|:---:|:---:|:---:|
| 1 | High | High | Medium-low |
| 2 | High | Low | High |
| 3 | Low | Low | Medium-low |
| 4 | Low | High | Low |

In other words, Proposition 1 implies that when the production sensitivity is high, and technology deployment is high, prior literature predicts medium to low food wastage. Similarly, proposition 2 predicts that when the production sensitivity is high and technology deployment is low, there might be high food wastage in the supply chain. Propositions 3 and 4 can be interpreted in a similar fashion. These propositions were tested on the data gathered from the interviews and the prepared case-studies.

## 3. Results

The actual food wastage in each firm's supply chain was compared with the expected food wastage predicted from the propositions. Table 4 below describes the result.

Based on a comparative analysis of the expected food loss (according to the afore-mentioned propositions) to the actual food loss incurred by the firms in the samples, the following results were found:

i.　Out of the 17 firms studied in the sample, 13 firms showed similar actual food loss as predicted in the propositions. A majority of the firms support the predicted propositions. Thus, the overall findings suggest that platforms can help prevent food wastage in supply chains.

ii.　Four companies, namely, C3, C10, C13, and C17, did not have actual loss as predicted by the propositions. The author studied them in further detail to understand the conditions in which technological platforms do not help in preventing food wastage, or are not required to avoid food loss. The following are the conclusions based on these four anomalous firms where the technological deployment did not show any relationship with the waste levels.

iii.　At C3, food loss levels are higher than expected despite having a high technological infrastructure in their supply chain. The respondent shared that the firm had not efficiently installed technologies, and there were "gaps [they] needed to fill". C3 was planning to initiate "process intensification" to explore how they could use "technological platforms to reduce food loss." The interviewee shared that they were "planning to use data from ERP to identify the areas of supply chain having higher wastage" and then "improve them for future".

iv.　At C10, actual food loss levels are lower than expected. The interviewee shared that "the highest chances of food loss are in the raw material storage, which requires temperature and humidity controls". The staff of C10 is trained to monitor the temperature and humidity regularly. "At the raw material warehouse, the likelihood of wastage is high." Hence, "the air-handling units [there] are automated". C10 also implements lean manufacturing, which ensures low wastage levels despite limited use of technology.

v.　At C13, the waste is mainly the residue left after the flavors have been extracted from spices. However, C13 has been able to convert this "waste" into a by-product. They found alternate use

of the residue as a "filler" in spice-blends. C13 sells the residue to spice mix companies. Hence, their wastage levels are very low, despite having a reasonably low technological infrastructure and high production sensitivity.

vi.    At C17, the food wastage levels are very low despite having limited technology and moderate production sensitivity. This low wastage level is because the company relies on manual observation and checks. However, the interviewee shared that "in future, implementation of technological platforms can reduce manual work, improve efficiency, and reduce errors".

**Table 4.** Result of actual food wastage in each firm's supply chain compared with the expected food wastage predicted from the propositions.

|  | PS | TR | Expected Food Wastage | Actual Food Wastage | Comments |
|---|---|---|---|---|---|
| C1 | 8 | 7 | Medium-low | Low | Food wastage reduced from 3–4% to 0.7% with tech. systems; major decrease in back-end |
| C2 | 8 | 5 | Medium-low | Medium | Raw material loss reduced by 20–30% with tech. systems |
| C3 | 6 | 4.5 | Medium-low | Medium-high | Process intensification with ERP helped reduce wastage overall, tech. platforms and wastage found to be unrelated |
| C4 | 6 | 3.5 | Medium | Medium-high | Automation and ERP reduced wastage |
| C5 | 6 | 4 | Medium | Medium-high | Wastage reduced through computerized palletization and other processes |
| C6 | 8 | 6 | Medium-low | Low | Yield improvement observed after tech. deployment |
| C7 | 3 | 3 | Low | Low | Tech. systems reduced manual error |
| C8 | 10 | 4 | High | High | Use of some tech. systems reduced wastage by 8–10% |
| C9 | 7 | 1 | High | High | The use of some tech. systems reduced the wastage of the final product, though raw material wastage is still high |
| C10 | 8 | 2.5 | Medium-high | Low | Tech. platforms and wastage found to be unrelated |
| C11 | 3 | 2 | Medium-low | Low | Marginal waste reduction observed after installing ERP |
| C12 | 7 | 5 | Medium-high | High | Raw material wastage decreased with automated air-conditioned storage |
| C13 | 6 | 1 | High | Low | Tech. platforms and wastage found to be unrelated |
| C14 | 3 | 1 | Medium-low | Low | Tech. supported cold-chains may improve the yield |
| C15 | 2 | 3 | Medium-low | Low | Automation reduced wastage in baking process |
| C16 | 2 | 2 | Medium-low | Medium-low | Loss is mostly due to physical damage and not production sensitivity |
| C17 | 4 | 0 | Medium-low | Low | Manual monitoring ensures low wastage in absence of tech. platforms. Tech. platforms and wastage found to be unrelated |

## 4. Discussion

The overall conclusion from this study is that technological platforms can play a role in reducing food wastage in supply chains. For most firms, supply chain complexity and perishability of the raw material and products can serve as useful indicators to identify the relevance of technology.

Further, this study reveals that technological platforms can help reduce food wastage in supply chains, both directly and indirectly. The following examples show the direct effect of technological and other technology platforms in food supply chains.

First, automated PLCs reduce the chances of manual error and process failures. They also enable firm-level monitoring of a range of process parameters. Second, ERP helps identify efficient routing systems to improve logistics networks. Third, inefficiencies in the procurement system can be resolved with extensive backward integration. Technological platforms discussed in this study can facilitate the monitoring and control of such integrated supply chains to reduce wastage further. Finally, technologies like M2M communication enable significantly better control of ambient conditions.

All these effects of technological platforms help in reducing food wastage in supply chains by enhancing operational visibility and process control. Moreover, this study also revealed several other indirect effects by which technological platforms can help reduce food wastage. For example, combinations of technologies like ERP and barcode readers enable the development of methods like "ready-make-discard". With such methods, retailers can identify and sell the earliest manufactured product unit. Such methods are crucial for supply chains like C8, where the products are highly perishable. This finding suggests that the scope of technological platforms in reducing food loss goes beyond the improvement of parameters like visibility, precision, and efficiency. Like in the case of C8, these systems can enable newer practices in supply chains to reduce loss of perishables. Also, technological platforms can help identify areas of high wastage. Thereby, firms can initiate efforts for process improvement, like in C3. Further, technological platforms can improve demand forecasting by connecting food manufacturers to retail stores or restaurants. For example, rapid demand fluctuations were a significant challenge at C4, which caters to the fast-food restaurant industry. Their product has low shelf life even under refrigerated conditions. With the help of internet-based technological platforms, C4 could communicate with their customers more efficiently. This implementation significantly reduced the food wastage in their supply chain. Finally, technological platforms can enable automation of certain processes. Although these processes may still require manual monitoring, they can reduce food wastage, as was noted in the C5.

Thus, technologies enable other processes that can indirectly reduce food wastage in supply chains. In sum, the study supports the proposition that technological platforms have the potential to influence food supply chains in a manner that would reduce wastage.

The findings from the present study are derived from qualitative data from one country. Although the author explained why India is an appropriate context for this study, it is worth acknowledging that the infrastructural challenges may be different in other parts of the world. Hence, future scholars can study the relationship between food wastage and technological platforms in other parts of the world such as South-East Asia and Sub-Saharan Africa where food wastage in supply chains is significantly high [2]. The author tried to make the sample as representative as possible, by incorporating firms of different industry segments, sizes, and location, there are limitations in the generalizability of the study owing to the qualitative nature of the data and analysis. Future research can study the relationship between technological platforms and the food wastage in supply chains quantitatively (for example, using linear regression and other related models). Further analysis can also reveal other the conditions under which technological platforms can be more or less feasible to firms, and the conditions under which they can be effective in reducing or preventing food wastage. Thus, this paper has opened up several avenues for future research.

## 5. Policy Implications

Reduced food wastage can potentially help in improving food security, reducing hunger, and malnutrition that are the critical issues in India and other developing economies. One purpose of this study was to generate evidence to support the potential of technological platforms in food wastage management in developing countries. Collaborative and synergistic actions are from the government and private sector to implement policies that can address the high volumes of food wastage in supply chains. Improving supply chains in the processed food industry can make a substantial difference in developing economies, as most wastage occurs after harvest, but before the produce or end-product reaches the final consumer. This study identifies a range of technological platforms that not only appear promising to address this problem but have proved their effectiveness in several food-processing companies in India. The investments made for the installation of such technologies can potentially be amortized with the savings from prevented food wastage in the long-run. The findings from this study can be applied to other developing economies that suffer from high food wastage and poor technological infrastructure in supply chains to address hunger and food security concerns across the world.

**Funding:** This research received no external funding.

**Acknowledgments:** This study was possible because of the time and support by the interviewees who shared their valuable insights. The professors at the University of North Carolina (USA), Symbiosis International University (India), and Berlin School of Economics and Law (Germany) provided their suggestions towards the development of their research.

**Conflicts of Interest:** The author declare no conflict of interest.

## Appendix A. Interview Questionnaire

*Introduction before the interview*

Dear Sir/Ma'am,

My name is XXXX and I am doing research on food wastage in supply chains. With my research, I hope to better understand the gap in Indian industries leading to wastage and understand the infrastructure set up at companies of different operational scales. I will be very grateful if you could spare some time and answer this interview questionnaire. Please be assured that this thesis is for academic purpose and the information I gather will not be shared with any company. All individual and firm identities will remain confidential. Your response will be very valuable for my research. Thank you very much.

*About the interviewee:*

Name:

Designation:

Company name and Location:

*Part (A) Product and susceptibilities*

1. What are the products manufactured by your firm?
2. What is the scale of production in your company? Please specify whether this scale is for one facility and if there are multiple facilities owned by your firm.
3. Please describe the SC of your main products in brief. (The remaining questions can be answered specifically for one or two major products produced by your firm)
4. What is its shelf-life?
5. Are there any special requirements of your food product? (In terms of pressure/temperature/ humidity or storage)

      a.     At the time of production

      b.     At the time of transportation

6. How are these ensured during the processing stage/transportation?

7. How often is there a failure? (as an approximate estimate from your experience)

8. In case of a failure, how are the personnel notified? How is the corrective action then taken?

9. At what stage(s) of the production process/transport as the product susceptible to contamination?

10. What the ways by which any contamination can be detected? How is the personnel notified and corrective action taken?

*Part (B) About the raw material*

11. What are the raw materials used? What is their shelf life?

12. What the requirements of surrounding conditions (temperature, pressure, humidity)?

13. In case of a situation wherein the requirements are not met, how are the personnel notified? How is the corrective action then taken?

*Part (C) About the technological platforms and infrastructure*

14. What kind of technological platforms are set up in your firm? Since when?

15. Is any technology which uses integrated within the supply chain? For example: SAP, Enterprise Resource Planning etc. (Please specify which)

16. Are Dashboard Analytics and Control Tower used in any part of your supply chain? [Explanation if needed: Dashboard Analytics is an overview on performance of various parameters displaying summaries of different reports as widgets on a single page. It enables simultaneous monitoring of many metrics. Control Tower is a concept derived from airport management and now being used by logistics service providers. This technology comprises of a telecommunications tower, such as that for televisions, to monitor the movement of vehicles or goods.]

17. Is M2M communication used in the supply chain? [Explanation if needed: M2M refers to machine-to-machine communication. Here, a machine, such as a temperature sensor communicates about parameters like surrounding conditions or others. This information is communicated directly to the enterprise monitoring system where corrective action is taken in case of a failure without human involvement. M2M communication also has other applications like inventory replenishment and others, which need to be considered of this answer.]

18. If any of the above methods is used, since when it is used?

19. What were the key objectives with which these were installed? (E.g.: Improvement of productivity, efficiency, reduction of wastage etc.)

20. Was prevention of wastage a reason for which these systems were installed?

21. Has there been any change in product wastage because of these systems? (either in terms of percentage, quantity, volume or value)

*Part (D) About the wastage*

22. Are the wastages being tracked? What method is used?

23. What are the levels of wastage of food products (raw material or final product) in your company/in the immediate transportation? (In percentage terms or absolute figures, preferably for products requiring cold storage. If there are no such products, then for any other key raw material or product)

24. To what levels can these be related to absence of proper infrastructure (such as cold storage)?

25. In case proper technological facilities are available, does it so happen that due of certain fluctuation in the conditions, the quality of the product is hampered? Can it lead to a substantial wastage?

26. Do you see any correlation between implementation of technological information platforms and reduction in wastage?

**Appendix B. Case-Study of Each Sample Firm**

**B1. C1 case.** (Based on the interview with Assistant Manager, Sales)

The case study of C1 is based on information from both primary and secondary sources (company website and social media pages). This company is the market leader in India in the several FMCG sub-sectors with a valuation of USD 9 billion and annual turn-over of over USD 3 billion. C1 Ltd. has set up a model called "e-Choupal" through which they have been able to integrate the primary producers (farmers in remote villages) with their supply chains by means of information technology (IT), thereby revolutionizing agriculture in places they have penetrated. (Krajewski, Ritzman, & Malhotra, 2008, pp. 20–21). Since the model is very relevant to the thesis topic, further secondary research was done.

**Product range:** Food products: *Atta* and variants such as Multigrain *Atta*, Salt, RTE meals, biscuits, chips, sugar candies, and spice mixes.

4 *Atta*: *Atta* in Hindi means flour, or more commonly, whole wheat flour. In South Asia, typically in India, households normally buy *Atta* or flour and prepare the *rotis*, *parathas*, or other traditional flatbreads at home, unlike in the west where bread is commonly bought. Flour is consumed at homes in large quantities almost everywhere in India.

**Key challenges:** C1 relies largely on agricultural output. Due to the existing practices in agri-business and the presence of large number of middlemen involved, the quality of raw material tends to deteriorate before it reaches the food processor. The agricultural infrastructure in India is not developed in most parts. The commodities market in India is based on the village *mandi* system, which does not monitor the quality of the produce. Often, due to inefficient communication and transportation, large quantities of produce rot in the supply chain. (Krajewski, Ritzman, & Malhotra, 2008, pp. 20–21).

*Mandi* refers to a marketplace in villages and towns where the farmers sell their produce at a price decided on the supply, demand, and seasonal patterns. *Choupal*: is derived from the Hindi word "*Choupal*", which refers to a traditional gathering place in villages. As in the case of communications technology, the prefix "e-" refers to electronic means of communication.

**Procurement:** C1 has largely overcome the above-mentioned problems with the Choupal model set up in 1999. This model combines a web-portal in the local language (India has over 21 recognized languages and other regional languages) and personal computers with Internet access placed in the villages. This creates a communication channel between C1 and villagers. The produce is also collected from the farmers here. Information on prices of crops at different *mandis,* tips on farming practices, and so forth are also provided. This model has triggered higher payment based on produce quality and higher productivity. Choupal services today reach over 3.5 million farmers in over 33,000 villages across 6 states. (Krajewski, Ritzman, & Malhotra, 2008, pp. 20–21). Choupal provides means to collect the produce closer to the farmer with faster transportation and less wastage. (Neggehalli & Shankaran, 2008). The rest of the case is prepared based on interview.

**Scale of production:** *Atta*: 60,000 tonnes per month (TPM); biscuits: 20,000 TPM; Chips: 3500 TPM. Total pan-India production: ~85,000 TPM.

**Production and Distribution:** Procurement through Choupal is described already. Further supply chain is described for a single product (*Atta*), in order to have brevity in the interview and case study. However, findings of other areas, such as IT implementation and food loss, are applicable to all food products.

Quality checks for the raw material (RM) are done followed by grading, blending, and cleaning. Water-treatment softens the grains before grinding, which is done in three stages. This is checked by sieves, metal detectors, and microbial tests. Once packed, it is sent to warehouses, then to distributors, and then retailers.

**Key features of the RM:** Wheat and other grains like soya bean, gram lentil, oat, and maize; sodium bicarbonate, ammonium bicarbonate, sugar, salt; vegetables, oils, lentils, and spices are the major raw

materials. These have a shelf-life of 6–8 months. Relative humidity (RH) <60% is needed. Vegetables need refrigeration (used in 2–3 days).

**Key features of the product:** Shelf-life varies from 3 to 12 months. All products are shelf-stable at ordinary temperatures. RH must not exceed 70%.

Shelf-stability implies that the product does not lose its quality, does not degrade microbiologically or otherwise, and does not require refrigeration until after opening (USDA, 2014).

**Requirements of the production process and transportation:** Temperature controls are required for almost every product during production.

**Failure occurrences and chances of contamination which could lead to food loss:** Process fluctuation beyond control levels occurs once in 2 months or less.

**Features in the supply chain/production process to prevent/correct failure:** RM is checked by QA department before usage. Sieves and metal detectors check the presence of contaminants. Temperature controls are automated using Programmable Logic Controller (PLC) set manually before every shift. If the failure still occurs, a signal notifies the personnel. WIP and FG are checked by QA.

**Food loss:** Wastage is tracked by the systems used to track production and distribution (SAP for production, Astra and Sify for sales). Production loss is ~0.7% of the production volume. Distribution loss account to ~2% of the production volume.

**Technological infrastructure:** E-choupal is used for procurement activities, which have their own server. SAP is used for production and logistics. For sales records and planning, "Sify" records data from distributor to retailer and "Astra" records date till the distributor. Dashboard Analytics is used in the Sales Process. GPS tracking is used for high value products like RTE meals, cigarettes, and personal care products. RFID is used till distributor warehouse and its data is fed directly into Sify and Astra. SAP was installed to coordinate the production with sales all over India, which is crucial, as there are over 1000 SKUs in food business alone. Due to SAP, supply chain efficiency and visibility have improved. Costs are optimized.

**Correlation between Food Loss and IT platforms:** Prevention of wastage was one of the reasons for installing IT systems. Food wastage in factory was ~3–4% earlier, now reduced 0.7% with IT integration. Marketing and distribution wastage has also dropped significantly, and there have been fewer product recalls. A major drop is wastage has been in the back end supply chain (from farmer to factory) with Choupal. Thus, there is a very strong correlation in reduction of wastage and implementation of IT platforms.

**B2. C2 case** (Based on the interview with Quality Assurance Officer, Operations)

In the present case study, the prepared dishes and cooking category sold under the brand name of Maggi, manufactured in Goa, are studied. C2 has 8 factories in India.

**Product range and scale of production:** Noodles, soups (premix), and tomato ketchup. Combined production volume is 252,060 TPM in this factory alone.

**Procurement, production, and distribution:** (Described for noodles) RM undergoes quality checks. Flour tipping is done automatically. Sieve testing is done. RM storage is avoided. Mixing and blending are the further steps. Dough formation is done in mixers. This is followed by pressurized steaming at specific temperatures. Cutting into noodle form and drying are done subsequently. Noodles are then fried and packed. Packed goods are sent to distribution centers. They are sent to distributors and from there to retailers.

**Key features of RM:** Most ingredients are shelf-stable at ordinary conditions. Tomato puree and dehydrated vegetables require refrigeration but are consumed soon on arrival.

**Key features of product:** Shelf-stable at ordinary conditions for >6 months.

**Requirements of production process and transportation**: Fryer and dryer require temperature controls (120 °C).

**Failure occurrences which could lead to food loss:** Failure occurrence was once in a year and half (based on interviewee's term of work).

**Features to prevent/correct failure**: In case of a failure, the entire line gets locked automatically and production is halted until maintenance is done. Quality and adulteration are checked by sieve testing, microbial testing, and metal detectors.

**Food loss:** Wheat flour: 0%; spices: 5–10%; puree: ~5%; final product: 1–2%.

**IT Infrastructure:** SAP is used. C2 has its own cloud-based server located in Australia. Data is shared by means of SAP at country-level on a regular basis. RFID is used for product tracking. Based on the interviewee's knowledge, M2M communication is likely to be used for sensitive products like milk-based pediatric nutritional supplement. In Europe and USA, C2 is likely to use extensive GPS tracking.

**Correlation between food loss and IT platforms:** Reduction in wastage is seen with better process regulation systems. Cloud-based information platforms can reduce wastage and provide other benefits like improved efficiency, higher production, and reduced labor requirement. For products like dehydrated vegetables, losses have reduced by 20–30% of the previous quantity of wastage prior to IT systems' installation.

**B3. C3** (based on the interview with Assistant Manager, Research and Development)

**Product range and scale of production:** Milk-based nutritional drink, energy drink, tomato ketchup, and Sampriti *ghee* are the products. Pan-India production: milk-based energy drink: 28,600 TPM; energy drink: 22,500 TPM; ketchup: 2500 TPM. Total production: ~53,600 TPM. *Ghee* is clarified butter commonly consumed in households in the Indian subcontinent.

**Procurement, production, and distribution:** This is explained for the milk-based energy drink. Milk is sourced directly from suppliers, checked by QA, refrigerated, and used on the same day. Milk is converted into powder by spray/drum drying. Sugar is added, and caramelization is done by heating. Additives and oils are added and mixture is emulsified. Vitamins, minerals, antioxidants, flavors, and colors are added. Dry blending is done, followed by packaging and dispatch. Distribution is handled by carry-forwarding agents.

**Key features of RM:** Milk requires refrigeration. Additives, vitamin–mineral mix, flavors, are table for 3–6 months; tomato puree is aseptically packed (shelf-stable).

**Key features of product:** Shelf-stable for over 9 months under ordinary conditions.

**Requirements of production and transportation:** For the milk-based energy drink, spray/drum drying is to be done at ~220 °C, and RH is <55%. During packaging, the temperature is maintained below 25 °C. For ketchup, heating at ~80 °C is needed. Packaging requires sub-atmospheric pressure.

**Failure occurrences which could lead to food loss:** Once in ~3 months, the process fluctuation goes beyond acceptable levels. Minor fluctuations occur ~once a month. **Features to prevent/correct failure:** All temperature and pressure controls are digitally displayed and manually handled. Alarm rings in case of high fluctuation to notify operators. Production line is monitored every 30 min by the QA staff.

**Food loss:** Wastage is tracked using SAP. RM loss is 2–3%; WIP loss is 5–6%; post-production loss is <1%. Total food loss is ~9–10%.

**IT infrastructure:** SAP is used in production. PLC has a positive release system, i.e., only when products are fit (when tested by metal detector). "Viper" is used in QA. Dashboard analytics are used in QA. Barcodes are used post-dispatch until sales. Their data is integrated into SAP. M2M communication and Control Tower technology are not used.

**Correlation between food loss and IT platforms:** If food loss as seen in SAP are higher than usual, the process is redesigned using the SAP data with regard to areas of higher loss. This is called "process intensification". As an estimate, it can be said that the wastage reduction on implementation of SAP was ~5–8% of the initial value.

**B4. C4 case** (based on the interview with production manager)

**Product range:** C4 is the sole supplier to a pizza restaurant chain and a chain of cafeterias. Only dough balls are produced in plant. Vegetables are supplied directly to the restaurants.

**Scale of Production:** Pan-India supply of all food items: ~5000 TPM, but there is a large fluctuation based on festivals and promotional activities.

**Procurement, production and distribution:** Raw material (flour, gluten, yeast, oil) is loaded at set temperatures into the batch mixer. Slow mixing is followed by fast mixing for set durations. Dough is cut and rolled into balls of 20, 30, and 50 g. These are passed through metal detectors, packed, and transported to restaurants in cold chain vans.

**Key features of RM:** Yeast must be stored at <4 C, so that its activity is retained. Flour requires RH <60%. Other RM is fairly stable for 1–3 months when sealed. Inventory of 1 month is held to handle fluctuations. Vegetables are supplied daily.

**Key features of product:** Dough balls have a shelf-life of 5 days at 4 C.

**Requirements of production and transportation:** Temperature control (<28 C) is crucial for yeast activity. Transportation and storage at 4 C is required for dough balls.

**Failure occurrences which can lead to food loss:** Once in ~8 months, checked by QA.

**Features to prevent/correct failure:** The plant is largely automated. Temperatures of water outlet and yeast slurry are maintained automatically by PLC. Production information is captured by the globally connected server monitored by the regional production engineer.

**Food loss:** Tracked on SAP. Factory: ~2%. Restaurant wastage (by consumers) is high.

**IT infrastructure:** SAP provides global connectivity. At restaurants, the demand fluctuates significantly. This necessitates a well-integrated supply chain. The franchisee outlets are integrated in SAP system, which enables fine-tuning of forecast. Cloud-computing is used to share the information at regional and global levels. M2M communication is not used. Dashboard analytics and Control Tower are not in use but would probably be used in USA. Barcode and RFID are not used. Entries for distribution are made manually in SAP.

**Correlation between food loss and IT platforms:** Before automation and SAP, food loss was about 10–12% of total production. Now, it is 7–8%. IT infrastructure was set up to improve service levels and efficiency. The franchisee orders are very dynamic, and SAP improves forecasting, thereby reducing wastage. Losses are also reduced because of superior IT enabled controls.

**B5. C5 case** (Based on the interview with Quality Assurance Executive, Operations)

**Product range:** C5 Green and C5 Elephant beers are made in the studied production plant. **Scale of production:** ~170 million liters per month. Considering the density of beer to be 1.05 g/mL9, ~172 million TPM is the total production. (*Own calculation done as per industry standards, for company classification based on production scale in section*). Specific gravity refers to the mass occupied by a standard unit of volume expressed in different units. Source: (Manning, 1993).

**Procurement, production, and distribution:** RM, consisting of malt (fermented barley), rye, and hops is imported from Denmark and tested for microbial activity. Milling is done to dissolve starch at 38–50 °C followed by cooking at 65 – 70 °C to deactivate enzymes. Hops and yeasts are added to the supernatant liquid with simultaneous aeration to circulate controlled levels of oxygen. Sugar is converted to ethyl alcohol, and the process is tightly monitored. Bottling and carbonation are carried out, which are then followed by distribution.

**Key features of RM:** Shelf-life: 3–7 days, malt: protection for air and temperature <5 °C.

**Key features of product:** Shelf-life: 6 months.

**Requirements of production and transportation:** Many steps require strict temperature regulation. Bottling requires temperature and pressure control.

**Failure occurrences which can lead to food loss:** Fluctuations are very frequent. Manual corrections are done almost immediately. In a shift of 8 h, corrective steps have to be taken ~15–20 times. ~135 parameters need monitoring, done manually.

**Features to prevent/correct failure:** Temperatures are displayed and checked at intervals. Being a microbial process, not just a thermal and chemical one, these are very crucial for fermentation.

Positive release system holds back burst bottles. Parameters are monitored every hour. Filtration and product testing ensure absence of contamination.

**Food loss:** In factory: 4–5%; in warehouse: ~3%. Most wastage is attributed to process sensitivity. Fluctuations in condition are a major cause of failure and rejections.

**IT infrastructure:** The IT system is outsourced to Wipro. "Navizon" is used for production. Dashboard analytics is used in production, as 135 parameters need to be checked. M2M and Control Tower are not used. Barcodes are used in distribution. Data are shared manually with enterprise integrated with Navizon through cloud-based server.

**Correlation between food loss and IT platforms:** After the implementation of IT systems, higher levels of automation could be reached, like automated palletization controlled by an advanced IT system. Wastage reduction has resulted as an outcome of IT integration, although it was not a primary objective.

**B6. C6 case** (Based on the interview with Category Manager, science and technology)

**Products:** Chocolate brand1 and Chocolate brand2 (chocolate bar); Bubbaloo (chewing gum), creamy biscuits, milk-based drink; Tang (sherbet); candy; toffee.

**Scale of production**: Pan-India: chocolates: 5000 TPM; gums and candies: 2500 TPM; beverages: 2500 TPM; biscuits: 1250 TPM, Total: ~11,250 TPM.

**Procurement, production, and distribution:** A brief and generic process for chocolate production is described. Parameter and certain steps may vary for different varieties. Cocoa, the key RM, is procured from West Africa. Milk, sugar, and other RMs are sourced locally. Post-harvest fermentation is carried out at 18 °C. Roasting is done in furnaces (200 °C). This is followed by milling to separate cocoa liquor from cocoa butter. Heat treatment is given for flavor development. Blending is done with milk, sugar and flavor, followed by homogenization of the chocolate mass. While still liquid, chocolate is put into molds and gradually cooled. After QA testing and packaging, it is sent to stockists and then to retailers.

**Key features of RM:** Cocoa, once fermented, requires cold storage. Sugar requires dry surroundings. Milk requires cold storage and must be used on the same day.

**Key features of product:** Shelf-life: 5 months to one year under ordinary conditions.

**Requirements of production and transportation:** Beans must be stored at temperature <18 °C. RH must be 65–70% during fermentation at 18 C. Roasting temperature (~180 °C) is crucial. Different stages have different temperature requirements. RH should always be <50%. Reefer containers used for Chocolate brand1 Silk variant keep temperature <25 °C.

**Failure occurrence which can lead to food loss:** about once in 3–4 months or less.

**Features to prevent/correct failure:** Equipment has a hot water jacket for temperature maintenance. Temperature fluctuations are handled automatically. PLC displays the temperature. Metal detectors and sieves are used. Product is tested by QA.

**Food loss:** RM: ~1%; FG: 0.8%; pilferage: 1–2% of production volume.

**IT infrastructure:** Customized SAP is used in procurement, production, and distribution. Dashboard Analytics generates reports for processes and inventory. Control Tower is used at a global level. GPS based tracking is done in certain cases. IT systems were installed to increase productivity and efficiency. RFID is used in distribution.

**Correlation between food loss and IT platforms:** It was claimed by the interviewee that yield improvement had been observed after the implementation of IT systems, although it could not be quantified. The present yield is 98%.

**B7. C7 case** (Based on the interview with Corporate Trade and Marketing manager, distribution)

**Product range:** Several confectionery sweets.
**Scale of production:** Total 750 TPM in 3 facilities.

**Procurement, production, and distribution:** RM consists of sugar, flavors, mint extract, gelling agents, glucose syrup, and vegetable oil. Products like hard-boiled sugar candies undergo their processing, which includes the melting of sugar with glucose syrup and other ingredients. Temperature needs to be controlled to prevent caramelization. Flavor and color are mixed. Once the right consistency is obtained, the slurry is poured in molds, cooled, packed, and dispatched. Transportation is at ordinary conditions.

**Key features of RM:** Sugar is the key RM, which needs dry conditions for storage.

**Key features of product:** Shelf-life: ~1 year under ordinary conditions.

**Requirements of production and transportation:** Strict temperature control needed.

**Failure occurrence which can lead to food loss:** about once in 3–4 months or less.

**Features to prevent/correct failure:** Corrective actions for temperature are carried out automatically as the system is programmed. If the systems fail, the personnel are notified by an alarm. Pilferage and damage are the issues faced in transportation. No system is used to prevent these, as costs incurred in its monitoring could exceed loss. Heavy metal detection is done by metal detectors.

**Food loss:** Including pilferage: 9–12%. In plant loss: ~4–5%. Wastage due to absence of proper infrastructure (e.g., improper storage facility for sugar): ~2%.

**IT infrastructure:** SAP is used. Information about inventory levels is communicated on cloud. Dashboard analytics widgets are used in QA. Barcodes/RFID is not used. M2M communication is not used in India, but it is probably used in USA and Europe.

**Correlation between food loss and IT platforms:** Low levels of wastage can be attributed to the presence of automation. These systems reduce the scope of manual error and improve the speed of corrective action. The role of IT systems to support automation is a significant factor in reduction food loss.

**B8. C8 Ltd.** (based on the interview with Corporate Quality Auditor, production)

**Product range:** The products come under three categories:

Retort packed10: *dals* (lentil soup cooked with spices) and vegetable curries; frozen foods: *parathas*, *naans* (traditional Indian flatbreads), *samosas, tikkis* (Indian snacks), and similar products; Cook-&-Sell category: Mainly milk-based Indian sweets. Retort-packed foods are food products, which are hermetically sealed in flexible pouches for long-term unrefrigerated storage (Webster, Definition of Retort Pouch, 2014)

**Scale of production:** Retort: 90–120; TPM per product; frozen: ~150 TPM per product; Cook-&-sell: 120–180 TPM for each product. Total production: >2000 TPM.

**Procurement, production, and distribution:** RM is bought from suppliers who obtain it from the agricultural cooperatives. RM is checked by QA, segregated as per its usage in final product. Production processes are very diverse as all products are significantly different, e.g., preparation of *dals* or curries involves pressurized cooking in steam-jacketed vessels, and preparation of *parathas* involves dough making and roasting. When prepared, the products the products are packed, labelled, and gradually frozen to −1° C. The orders from restaurants are classified into four regions, each of which has a centralized kC1hen equipped with QA labs. After further processing at centralized kC1hens, products are segregated as restaurants' requirements. At the restaurant kC1hens, the products are heated, garnished, and served.

**Key features of RM:** Flour, spices, dairy products, vegetables, coffee beans, chocolate, fruits, lentils, and many others are required. Most RM must be used in 3–4 days of procurement. Refrigeration at <5 °C is required for certain RM.

**Key features of the product:** Shelf-life varies from 1–2 days to 6–7 days.

**Requirements of the production and transportation:** several temperature regulations are required at different stages. Frozen products need transportation at <5 °C.

**Failure occurrence which can lead to food loss:** As the product range is very wide, controls have to be set and reset repeatedly. Failure of temperature regulations occurs once in 3–4 weeks per product line.

**Features to prevent/correct failure:** Temperature during transportation is checked by means of a temperature probe. In production, if the temperature fluctuates beyond limits it is reset manually. Maintenance measures are carried out in case of failure. Product is checked under X-Ray, metal detectors, and by QA.

**Food loss:** Mainly raw vegetables get wasted. Retort category: salad: 30–40%; gravy: 20%; curries and *dals*: 20–30%; but this also includes unusable portion of the RM like peels. Unusable raw material is sold as cattle feed. Loss of usable material: ~10–12%

**IT Infrastructure:** Citrix and Navizon are used for inventory management and QA. Dashboard analytics, Control Tower, and M2M communication are not used. Barcodes are being used till dispatch form centralized kC1hens.

**Correlation between food loss and IT platforms:** ~8–10% wastage reduction has resulted from IT implementation, as it is possible to more accurately know exactly which lot arrived and accordingly use it within its shelf-life. A major contribution of these technologies to decrease wastage has been through the development of "Ready-make-discard" method. In this method, once the product is packed, it must be used within 4 days, and once reheated at restaurants, it must be served within 4 h. This information is conveyed using barcodes. The product can be served on priority without being wasted. This would not have been possible without IT integration.

**B9. C9 case** (based on interview with Quality Assurance executive, production)

**Product range and scale of production:** Ice-creams of over 20 flavors; 20 TPM total.

**Procurement, production, and distribution:** RM (milk, sugar, fruits, and dry fruits) is sourced from different vendors. Milk is pasteurized, tested by QA, and condensed using evaporators. Other RM is cleaned and tested. Spiral mixer is used to blend the ingredients and freeze the mixture to −20 °C. Simultaneously, aeration is done. Hardening of ice-cream is done post-packaging, by cooling it to −40 °C. Ice-cream is transported in boxes filled with solid carbon dioxide, which maintains the temperature < −18 °C even when transported by road to the farthest store.

**Key features of RM:** Milk: storage and transport <4 °C. Others: storage <5 °C; <2 days.

**Key features of product:** Below −18 °C, shelf-life: 2–3 weeks.

**Requirements of production and transportation:** Process controls described earlier.

**Failure occurrence which can lead to food loss:** Transportation failure: never occurred. Failure in the production process: often during installation.

**Features to prevent/correct failure:** Ice-cream is transported in boxes at −50 °C. Its temperature never exceeds −18 °C. Process failure correction is done manually.

**Food loss:** RM loss: 7–8%; final product: 2–4%.

**IT infrastructure:** SAP is used in production and distribution to facilitate coordination. Dashboard analytics, Control Tower, M2M communication, RFID, and barcode are not used.

**Correlation between food loss and IT platforms:** Earlier, product wastage was ~8–10% of production. Now, it is ~2.5%. There are fewer product recalls with better supply chain coordination. Better technological infrastructure can further reduce wastage.

**B10. C10** (based on interview with Production Executive)

**Product range and scale of production:** Flavors: liquid, powder, encapsulated powder, emulsion; liquid fragrances. Powdered flavors: 350 TPM; liquid: 80 TPM.

**Procurement, production, and distribution:** Production and procurement are based on orders. Production process cannot be described due to secrecy policy.

**Key features of RM:** Cardamom, clove, and so forth, and solvents like alcohol. Storage temperature must be <10 °C, RH, 45–60%; maintained by air handing units (AHUs).

**Key features of product:** High-value products having a shelf-life of 4–6 months.

**Requirements of production and transportation:** Different production operations are required for different products. Few products need air-conditioned vehicles.

**Failure occurrence can lead to food loss:** less than once a month.

**Features in supply chain to prevent/correct failure:** AHUs maintain the temperature and RH. HACCP and GMP are implemented. Metal detectors and Filtration membranes ensure absence of contamination. Sensory analysis by trained personnel and QA checks are done. Each machine has an operator who observes parameters on PLC screen and makes manual corrections if needed. Fluctuations in AHUs are handled automatically.

**Food loss:** Wastage is tracked manually, and lean manufacturing has recently been implemented. Wastage levels are kept very low (0.5–1% in spillage and total 2%).

**IT infrastructure:** SAP is used for information sharing globally. Purchase starts automatically from integrated IT setup on order receipt. 10 is likely to use Control Tower and M2M at global level, not in India. Barcodes are not yet used in India.

**Correlation between food loss and IT platforms:** SAP improves inventory control, lead-time, and logistics tracking, which impacts wastage levels. In case of higher production, possibly in the future, GPS tracking can be a useful tool to reduce loss.

**B11. C11 case** (based on interview with Production Executive)

**Product range, production scale:** Tea, coffee, and soup premixes; total: 120–130 TPM.

**Procurement, production, and distribution:** RM is sourced from suppliers through middlemen, checked, and graded on arrival. Tea leaves are withered by passing hot dry air (180 °C), which triggers biochemical reactions. The leaves are subjected to roll-breaking and fired by hot air at 210 °C. Black tea so formed is graded and blended. Packaging is done, and it is then transported to warehouses, then to end customers. Coffee undergoes roasting and has different operations and temperature controls.

**Key features of RM:** Tea and coffee must be stored in RH 40–60%.

**Key features of product:** Shelf-life: 9 months under ordinary conditions.

**Requirements of production and transportation:** Process parameters already described. Transportation is done under ordinary conditions.

**Features in supply chain to prevent/correct failure:** Dehumidifiers are present in warehouse and factory. Temperature and RH are checked manually every hour. In case of failure, plant is shut down and maintenance is done.

**Failure occurrence which can lead to food loss:** once in a year or less.

**Food loss:** RM: 0.2–0.6% of the total RM; in-storage: ~3%. FG: ~1%.

**IT infrastructure:** SAP is used for information related to production. BAAN and InforERP are used for functions like accounting and sales. RFID, M2M communication, dashboard analytics, and Control Tower are not used. Barcode is used in distribution.

**Correlation between food loss and IT platforms:** Reduction in food loss was not an objective for installation of IT systems. Marginal reduction in food loss has occurred due to IT implementation. However, in case of storage of RM, if better process controls are available, using automation supported by IT systems, loss can be further reduced.

**B12. C12 case** (Based on the interview with Quality Assurance Executive)

**Product range:** Semi-fried potato fries, potato wedges, and patties.

**Scale of production:** 2500 TPM (only one production facility in India).

**Procurement, production, and distribution:** RM procurement is done through contract farming channels. Potatoes are sorted and subjected to peeling (by steam), trimming, cutting, blanching (hot water treatment), drying, frying, and freezing. Product is packed, palletized, and dispatched in cold storage vans to restaurants and retail distributors.

**Key features of RM:** Temperature must be 7–8 °C to prevent spoilage.

**Key features of product:** Shelf-life: 2 years under refrigeration at 2–4 °C.

**Requirements of production and transportation:** Temperature control in heating operations is very critical and depends on sugar content found by quality checks.

**Failure occurrence which can lead to food loss:** Frequent temperature adjustments have to be made based on sugar content, hence high manual intervention is involved.

**Features to prevent/correct failure:** RM is stored in automated AHUs. Holding time is automated as per temperature set manually.

**Food loss:** Processing: 8–15%; transport: 4–6%. Total: 12–15. RM wastage tracked includes unusable portion (peels), hence loss of consumable portion cannot be specified.

**IT infrastructure:** SAP is used. Plants in Canada and USA use cloud logistics. AHUs (RM warehouses) use cloud-based system for temperature monitoring from office. Barcodes are used till distribution.

**Correlation between food loss and IT platforms:** High levels of food loss can be associated to some extent with poor IT integration and low automation in processing. RM wastage is largely reduced with automated air-conditioning. Automated temperature controls will be more efficient than manual ones.

**B13. C13 case.** (Based on the interview with Head of Production Planning)

**Product range:** Main products are oleoresins11 of black pepper and chili. Product portfolio includes spice oleoresins, spice oils, natural food colors, essential oils12, antioxidants, floral extracts, mint-based products, and other specialty products.

Oleoresin: A liquid or semi-liquid preparation extracted by solvent consisting of fixed or volatile oil holding resin in solution. (Collaborative International Dictionary of English). These are preferred by customers over ground spices, as they offer standardization, volume reduction, and longer shelf-life; Essential oil: volatile odoriferous oil that give plants their characteristic odors and other properties, obtained by steam distillation, expression, or extraction (Meriam Webster Dictionary, 2014)

**Scale of production:** Total production: black pepper and chili oleoresins: 500 TPM. Other products are made as per demand in much smaller quantities.

**Procurement, production, and distribution:** Procurement is done from different parts of the world all around the year. Once the RM reaches the factory, it is tested by QA. De-dusting prevents contamination. Flaking and grinding increases the surface area for extraction. This is followed by steam distillation (105 °C). The mixture of steam and volatile oils is condensed, and volatile oils are separated by sedimentation. The residue subjected to solvent extraction. The resinous matter responsible for the taste is extracted, and solvent is separated by evaporation. It is then packed and then stored until dispatch.

**Key features of RM:** Shelf-life: 5–7 months. It is stored in ventilated quarantine.

**Key features of product:** Flavor properties are retained for 6 months. RH <70%.

**Requirements of production and transportation:** Distillation: 105 °C; condensation: ~20 °C. Transportation is done under ordinary conditions.

**Failure occurrence which can lead to food loss:** Failure is avoided by regular monitoring. Contamination is very unlikely, as the process is sealed.

**Features to prevent/correct failure:** Monitoring gauges are provided on each reactor. Equipment is flame-proof. Filtration ensures absence of impurity. QA testing is done.

**Food loss:** Food wastages are not being tracked, because it is a zero-wastage production system. Solvents are reused. Residue is pulverized and sold as inferior grade spice powder to companies. Loss is <0.5% of production volume.

**IT infrastructure:** SAP is used for production and supply planning done jointly from offices in different cities, hence a cloud-based system is used. RFID, barcode, dashboard analytics, Control Tower, and M2M are not used. Data are shared manually.

**Correlation between food loss and IT platforms:** There is a correlation, but in this industry, it is applicable more in terms of the RM supply planning, which needs to be fine-tuned, as RM prices fluctuate. In cases of improper planning, there would be a large amount of wastage. Losses are indirectly reduced by SAP through better planning.

**B14. C14** (Based on the interview with Head of Supply Chain Management)

**Product range:** Nutraceutical1 range (lutein extract-based products) and Nutraceutical 2 (capsaicin extract-based ingredients), sold in the form of beads and oil suspension.

**Scale of production:** low volume high value product; ~2.2 TPM.

**Procurement, production, and distribution:** Procurement is done for the RM requirement of the entire year, as RM is seasonal. Marigold (source of lutein) plantation is done by contract farming. On harvesting, the flowers are allowed to ferment in cement pits and converted into pellets. These are subjected to saponification (conversion to suitable chemical form), extraction, solvent separation, and drying. These are strictly controlled chemical engineering operations required to obtain the required composition of product. Packaging is done, followed by transportation.

**Key features of RM:** Pellets need to be stored at −10 °C to prevent lutein breakdown.

**Key features of product:** Shelf-stable for 12 months under ordinary conditions.

**Requirements of production and transportation:** Strict temperature and solvent checking.

**Failure occurrence which can lead to food loss:** once in two months or less.

**Features to prevent/correct failure:** Data logger is provided for operators to record temperatures at intervals. Testing is done by QA.

**Food loss:** <1% as the inferior quality residue is sold as by-product.

**IT infrastructure:** ERP is used, but not for extensive data sharing. M2M, dashboard analytics, Control Tower, barcode, RFID, and PLC are not used. Tracking is done manually.

**Correlation between food loss and IT platforms:** The current production and wastage level neither demand nor can justify the expenditure of extensive IT integration. However, if cold-chain transportation is maintained for pellets and other raw material, less lutein would be lost in by-product streams. This would need IT systems' support.

**B15. C15 case** (Based on the interview with Quality Assurance Executive)

**Product range:** Biscuits.

**Scale of production:** ~2.2 TPM collectively.

**Procurement, production, and distribution:** RM is checked and stored till use. Wheat flour, oil, and sugar syrup are first mixed, followed by addition of soya lecithin, flavor, and sugar. Common salt, sodium bicarbonate, and ammonium bicarbonate are mixed with cold water and added. Quantities are controlled by PLC. Dough making is done, followed by cutting and baking. Natural cooling is done before packaging and shipment.

**Key features of RM:** Shelf-life: 6 months under ordinary conditions; sugar-dry storage.

**Key features of product:** Shelf-life: 9 months under ordinary conditions.

**Requirements of production process and transportation:** Stringent temperature controls are required for baking. Controlled quantities of flour and ingredients to be fed.

**Failure occurrence which can lead to food loss:** About once in 3–4 months.

**Features to prevent/correct failure:** Daily checks are done by operators. Temperature fluctuations up to 5 °C are handled automatically. Beyond this limit, manual interference is needed. Burnt and undercooked biscuits are manually removed after visual inspection. Contamination is checked by metal detectors and QA testing of final product.

**Food loss:** Broken and undercooked biscuits are recycled in dough making stage. Only burnt biscuits constitute food loss. Total loss is ~2% of total production volume.

**IT infrastructure:** ERP and Data Logger are connected to PLC. ERP setup is used to communicate with the Parle (client company for whom the production is done). Dashboard analytics, M2M, Control Tower and RFID are not used. Barcodes are used.

**Correlation between food loss and IT platforms:** Cannot be answered, as extensive IT infrastructure is not used. However, with automation in baking process control, improvements in wastage levels were seen.

**B16. C16** (Based on the interview with Head of Production)

**Product range:** *Chyawanprash* and *Asav Arisht* (health supplements based on traditional Indian formulations).
**Scale of production:** *Asav Arisht*: 500,000 liters per day; *Chyawanprash:* 10,000 tonnes per day; total production: ~316,500 TPM.
**Procurement, production, and distribution:** Sourcing is done from multiple vendors. *Amla* fruit is cleaned; herbal decoction is prepared. Pulp is extracted, followed by frying of pulp in *ghee.* Other herbal ingredients are added. It is further cooked, packed, and dispatched. *Asav Arisht* is prepared in the form of a decoction of several herbs and their extracts. Boiling at about 110 °C is done under a closely monitored environment.
Amla: Fruit of *Phyllantus emblica officinalis* tree, known for its very high vitamin content and health promoting properties (Emami Interviewee, 2014).
**Key features of RM:** Shelf life: 0.5–5 years depending on RM; *Amla:* ~2 weeks.
**Key features of product:** Shelf-life is 2–5 years.
**Requirements of production and transportation:** Temperature control during cooking; transportation is done under ordinary conditions, as products are shelf-stable.
**Failure occurrence which can lead to food loss:** Never observed by the interviewee.
**Features to prevent/correct failure**: Certified quality auditors make regular checks. Gamma radiation is used for microbial decontamination. QA checks are also done.
**Food loss:** RM:~4%; FG: 0.5%. Loss mostly due to packaging material (glass bottles).
**IT infrastructure:** SAP is used for delivery, payment, workers' attendance, production, inventory, and logistics tracking. Barcode is used in distribution.
**Correlation between food loss and IT platforms:** A weak correlation was suggested.

**B17. C17 case.** (based on interview with Senior Manager, production)

**Product range:** tablets, capsule, syrup, paste, powder and ointments under functional foods/ nutraceuticals/dietary products, such as *Chyawanprash.*
**Scale of production:** Total production: ~135 TPM.
**Procurement, production, and distribution:** Raw material is sourced locally. RM is mostly herbs, and herbal extracts are already in dry form. The RM is processed as per the SOPs and undergoes tableting, encapsulation, mixing, or other processes. These are then sent to the customers (companies like Hindustan Unilever, or hospitals).
**Key features of RM:** Shelf-life of ~1 year, as herbs and extracts are already in dry form.
**Key features of product:** Shelf-stable for 2–3 years under ordinary conditions.
**Requirements of production and transportation:** Different products have different process techniques and process control parameters, e.g., for *chyawanprash,* pulp must be boiled at 130 °C and RH: 35–50%. Transportation is under ordinary conditions.
**Failure occurrence which can lead to food loss:** Process failure has never occurred in the experience of the interviewee. Manual setting of process parameters is done regularly. Contamination, possible during RM dispensing, is prevented by QA checks.
**Features to prevent/correct failure:** Microbial Analysis is carried out manually, for the RM and final product. HVAC (heating, ventilation, and air-condition) of the production is outsourced, and the contractor performs regular checks.
**Food loss:** RM: 0.4 – 0.5%, Final Product 2.0% of the production volume.
**IT infrastructure:** No IT platforms in supply chain. Only Tally is used for accounting.
**Correlation between food loss and IT platforms:** No correlation is drawn between food loss and IT platforms by the interviewee based on her experience and knowledge of the company. The wastage levels are fairly low without the deployment of any IT based technology, which can be attributed to regular manual monitoring by trained staff.

On the basis of the above case summaries, inferences were drawn regarding role of cloud-based information platforms to reduce wastage in food supply chains, which is presented in the subsequent sections.

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
