# Peer review of "Food Waste Management with Technological Platforms: Evidence from Indian Food Supply Chains"

_sustainability, doi:10.3390/su12198162_

Round 1

Reviewer 1 Report

I thank you for the opportunity to review this paper on the impact of technology in the process of reducing food wastage. The topic is timely and the literature on food waste problems has burgeoned in recent years, which means that there is a need for papers that propose a new approach to exploring knowledge in this area. However, in its current stage, the manuscript is not well-developed, clear, and convincing enough to justify publication.  Please see my detailed comments below. Good luck with your research!

In my opinion, the paper will be stronger if the author considers the following remarks:

Title

First of all, the title is not fully understood and may be misleading: Does the work concern issues related to the food system or food waste management? This is not the same.

Abstract

After reading a title and abstract, it isn’t clear to me what the purpose of this article is? Additionally, in the abstract, I did not find clear information about methods, results, and conclusion.

Introduction

Section 1.1. and 1.2. . in my opinion, the information presented here should be included in the materials and methods section.

Lines 24-37 - I suggest to provide current data on the scale of food waste.

Line 30 - authors wrote "... In most developed countries, food wastage mainly occurs at the consumer end" - I suggest to confirm it with appropriate citations.

Line 33 - incomprehensible transition to information about India; why only data on fruit and vegetable losses were provided?

Line 42 - authors wrote "... researchers ..." but reference only one paper? why?

Line 46 - authors wrote "... While research shows…” - what research?

Line 60 - authors wrote "... I identified ..." - explain how it was done, and specify the selection criteria

Line 62 - authors wrote "... representative sample ..." - explain how was sample representativity established?

Section 1.2. - explain in more detail how the technology platforms were identified? Specify the selection criteria. Why is quite old literature cited?

In general, the introduction is, in my opinion, quite chaotic, the aim of the work is not clearly indicated and the theoretical recognition of the topic is quite poor. The introduction doesn't build a logical case and context for the problem statement. The problem statement isn't clear and well-articulated. The conceptual framework isn't explicit and justified. The research question/s isn't presented.

I suggest that you consider a strong refinement of this section.

Materials and Methods

Lines 98-104 and lines 106-113 - identical content

Lines 99-100 - explain the reasons for the choice in more detail

The author wrote about the interviews: what were the questions? it is difficult to assess the correctness of analysis and inference without precise information about the research process.

The research design isn't defined and clearly described, and isn't sufficiently detailed to permit the study to replicate. The design and conduct of the study aren't plausible.

Section 2.3. The information contained in this part is incomprehensible to me, it is not clear what it stems from. What analysis does the author mean?

I suggest that you consider a strong refinement of this section.

Results

The information contained in this part is incomprehensible to me. It is not clear where these results came from.

How were food wastage levels classified and interpreted?

Line 152-153 authors wrote "... operational complexity and perishability, ... production sensitivity ..." - does not provide detailed information about these parameters? what do they mean how were they measured? very incomprehensible part of the article.

Line 163 - a detailed study of the selected companies has not been shown.

Line 165 - Why was such a choice made?

I suggest that you consider a strong refinement of this section.

Discussion and policy implications

Due to the lack of scientific rigor that occurred in the previous sections, I cannot refer to the information presented in sections 4 and 5 and indicate ways to improve it.

I suggest adding information on study limitations and future research and section "conclusions".

References

Many references have incomplete bibliographic data (e.g. lines 250, 251,255, etc.)

Errors in  references numbering (lines 261-269 etc.)

The number of references is quite small (only 27 items), additionally it isn't up to date (only 6  references from the last 5 years).

3 references are not cited in the text of the article (lines 256, 259, 274)

Reviewer 2 Report

The paper has to be improved because in this current form it seems to be more a technical report than an academic paper. Indeed, the references are not updated and they are scarce; there is often not care of editing; there are too many bulleted lists; poor critical approach. The abstract has to be revised as well as the introduction.

These are the main and general suggestions.

Details are reported in the file in attachment.

Round 2

Reviewer 1 Report

The author has addressed all the comments and suggestions I made in the first review. The quality of the article has significantly improved. In my opinion, as it stands, it adds new value to the literature already available on the subject.

I'm glad you found my feedback helpful to you. Best of luck.

Author Response

Dear Reviewer 1, 

Thank you so much for accepting my revisions. Yes, your comments were very helpful - not just for this present paper but in general for my overall writing improvement. I will keep your suggestions in mind as a continue doing research and preparing manuscripts for future. 

Once again, thanks for your comments. I really appreciate it!

Sincerely,

Youthika Chauhan

Reviewer 2 Report

The paper has mostly followed the suggestions. Surely, now it presents a more scientific approach. Nevertheless, there are still some inaccuracies to be corrected for publication.

References

Please, pay real attention to the editing and the information to be included. The references have not been standardized yet.

Line 39. I should include the main reference of FAO, because you mentioned it.

Line 41. The references [11] seems to be not in compliance with the topic of the sentence.

Line 994. Delete 14

Line 997. Delete 16

Line 999. Retrieved from? You use a new way of expressing the link of the document.

Line 1002. Delete 19

Line 1006. Available at? You use this specification only for some references.

Line 1009. Delete 23, as part of 1008.

Line 1014. More information. No sense reference, because there is only the name of authors.

Line 1037. There are missing pages and volume.

General inaccuracies

Line 59. Avoid the word “Conclusions”: it could be misunderstood as section of the paper. Instead, you decided to substitute “Conclusions” with “Discussion” and to add as final part “Policy implications”.

Line 62. Delete double point.

Line 79-80. You used correctly the author, instead of “I”. After, you used “we”. There is not correct connection. “We” is plural. Try to use impersonal form, if you do not repeat “the author”.

Line, 51, 91, 129, 134, 191, 225, 226, etc. You used “we”.

Line 166. Delete double point.

Line 170. You left “I”.

Line 219-221. Please, correct editing!

Line 233. Maybe, it is missing a verb. Incomplete sentence.

Line 251. You have already mentioned the acronym in line 225. So, please write the acronym.

Line 271-274. Food loss or food losses, please standardize.

Line 305-320. Cloud or technological?

Line 311 – 336. As I have already stressed in the first revision, there are too much bulleted list, above all one after another.

Author Response

Dear Reviewer 2,

Thank you very much for reviewing my paper and pointing out the inaccuracies. I realized that I had accidentally copy-pasted a wrong version of the references. The references were in alphabetical order rather than in the order of appearance in the text. I apologize for this mistake. I made the change in the order and also in the inconsistencies as you pointed out. I made other changes in the body of the manuscript as you pointed out. In the attached word document, I have described the correction as I made it in the revised version of the manuscript. 

I am also uploading a revised version of my manuscript. I believe this version should be OK as I have addressed all your points. I am looking forward to hearing your reviews. Once again, thank you very much. 

Sincerely, 

Youthika Chauhan
